# Practical N-to-C peptide synthesis with minimal protecting groups

Toshifumi Tatsumi[1], Koki Sasamoto[1], Takuya Matsumoto [1], Ryo Hirano[1], Kazuki Oikawa[1], Masato Nakano[2], Masaru Yoshida[2], Kounosuke Oisaki [1,2✉] & Motomu Kanai [1✉]

Accessible drug modalities have continued to increase in number in recent years. Peptides play a central role as pharmaceuticals and biomaterials in these new drug modalities. Although traditional peptide synthesis using chain-elongation from C- to N-terminus is reliable, it produces large quantities of chemical waste derived from protecting groups and condensation reagents, which place a heavy burden on the environment. Here we report an alternative N-to-C elongation strategy utilizing catalytic peptide thioacid formation and oxidative peptide bond formation with main chain-unprotected amino acids under aerobic conditions. This method is applicable to both iterative peptide couplings and convergent fragment couplings without requiring elaborate condensation reagents and protecting group manipulations. A recyclable N-hydroxy pyridone additive effectively suppresses epimerization at the elongating chain. We demonstrate the practicality of this method by showcasing a straightforward synthesis of the nonapeptide DSIP. This method further opens the door to clean and atom-efficient peptide synthesis.

[1] Graduate School of Pharmaceutical Sciences, The University of Tokyo, 7-3-1 Hongo, Bunkyo-ku, Tokyo 113-0033, Japan. [2] Interdisciplinary Research Center for Catalytic Chemistry (IRC3), National Institute of Advanced Industrial Science and Technology (AIST), Tsukuba Central 5-2, 1-1-1 Higashi, Tsukuba, Ibaraki 305-8565, Japan. ✉email: k.oisaki@aist.go.jp; kanai@mol.f.u-tokyo.ac.jp

Amide bonds are a recurring structural motif found in both naturally occurring and man-made molecules, as ubiquitously observed in the backbone of proteins/peptides, drugs, and functional materials. Amide bond formation is the most frequently used chemical reaction in medicinal chemistry; approximately 50% of drug discovery papers contain amide bond formation, twice as many as 30 years prior[1,2]. In recent years, efficient amide bond formations are especially important due to the emergence of medium-sized peptide drugs, which frequently exhibit unique characteristics and advantages over small molecule drugs and antibodies[3,4].

The traditional peptide synthesis iteratively elongates the chain from the C-terminus to N-terminus (C-to-N) using excess N-carbamate-protected amino acids and condensation reagents to minimize epimerization (Fig. 1a)[5]. Combined with solid-phase synthesis, the C-to-N elongation method has enabled facile construction of peptides of up to ca. 50 amino acid residues, and the introduction of revolutionary automated flow systems promises to increase this number[6]. Further, combined with native chemical ligation methods, synthesis of proteins with even 400 or more amino acid residues is also possible[7]. The maximum size of peptides/proteins that can be produced by chemical synthesis is rapidly increasing. Despite high fidelity and reliability, every C-to-N peptide bond formation requires multiple protecting-group manipulations and non-recoverable condensation reagents that produce waste. For example, the average molecular weight of an amino acid is ca. 110, but the molecular weights of commonly used protecting groups (Cbz: 135, Boc: 101, Fmoc: 223) or condensation reagents (EDC-HCl: 191, HATU: 380, COMU: 428, BOP: 442) are comparable to or much greater than the substrate. Furthermore, C-to-N elongation of anything longer than dipeptides often suffers from diketopiperadine formation, when a simple C-terminus ester protecting group is used[8]. Therefore, traditional peptide synthesis is of low atom efficiency and high environmental impact[9–12]. In an era more conscious of environmental preservation and sustainability, greener peptide synthesis is in high demand. Pursuing this, nonclassical amide bond formations[13,14] have been extensively studied, and some of them have been applied to C-to-N oligopeptide synthesis[15–23]. However, many protocols still require harsh conditions (high temperatures for azeotropic removal of water), super-stoichiometric reagents of sometimes poor accessibility, and protecting groups.

N-to-C peptide elongation (Fig. 1b) is less explored than C-to-N elongation due to difficulty in suppressing epimerization of the C-terminus amino acid residue's stereocenter[24–32]. Because the two amino groups, one in the elongating peptide strand and the other in the amino acid to be introduced, are already differentiated as amide and amine groups, respectively, this strategy is potentially advantageous in improving both atom and step efficiency by minimizing protecting group manipulations. Here we report an iterative and practical N-to-C peptide synthesis in liquid phase, in which epimerization is minimal. This method enables the use of unprotected amino acids and does not require elaborate condensation reagents, thus markedly improving the atom and step efficiency of liquid-phase peptide synthesis. Moreover, this method is applicable to convergent fragment coupling, as demonstrated in the short and scalable synthesis of a bioactive nonapeptide.

## Results and discussion

**Optimization of conditions**. To realize N-to-C peptide synthesis, we employed the peptide thiocarboxylic acid (PTC) platform (Fig. 1b)[33]. Amide bond formation using PTC under various conditions has been reported[34–43], but PTCs have never been used in iterative N-to-C peptide synthesis. Based on our previous development of a general, one-step PTC synthesis from peptides using a catalytic diacetyl sulfide (Ac$_2$S) and potassium thioacetate (AcSK)[44], we envisioned the scheme shown in Fig. 1b. Converting the C-terminus carboxylic acid to PTC differentiates the elongating peptide strand from the amino acid to be introduced without protecting groups. After peptide bond formation, the new C-terminus carboxylic acid can be directly used for the PTC formation to start the next elongation cycle. Elemental sulfur and water are the only waste produced in this peptide bond formation step.

We started optimizing the PTC-based N-to-C elongation using **1a** to produce tripeptide **2aa** (Table 1). Although PTC is inert to amide formation by itself, oxidatively dimerized diacyl disulfide is the active acylating species[45]. We first searched for aerobic conditions to convert **1a** to diacyl disulfides in situ, which would be captured by alanine calcium salt (Ca(Ala)$_2$)[46]. Using an iron(II) phthalocyanine (FePc) catalyst in open-air DMF, tripeptide **2aa** was obtained in 22% yield (entry 1). Next, N-hydroxy amine/amide/imide additives were screened to improve the reactivity while maintaining the low epimerization level (entries 2–6)[37]. Among the additives tested, 3-hydroxy-1,2,3-benzotriazin-4(3H)-one (HOOBt) afforded acceptable results, producing **2aa** in 33% yield and <1% epimerization (entry 6). When the concentration was increased to 100 mM, yield improved to 68%, while epimerization remained suppressed (<1% epi. level, entry 7). Then, the same conditions as in entry 7 were applied to the more sterically hindered dipeptide **1b**. Product tripeptide **2ba**, however, was produced only in a low yield (35%) with an increased epimerization level (6.1% epi. level, entry 8). HPLC analysis revealed that FcPc degraded the diacyl disulfide intermediate derived from **1b** prior to condensation. Therefore, we eliminated FePc, resulting in improved yield (64%, entry 9), although the reaction was sluggish (22 h) and the epimerization level was still high (4.9% epi. level). When alanine (H-Ala-OH), instead of Ca(Ala)$_2$, was used to mitigate basicity of the reaction system, the epimerization level was reduced to 1.3% (entry 10). Using DMSO as a solvent to promote diacyl disulfide formation[47], yield improved (70%) but the epimerization level increased (4.5% epi. level, entry 11). The epimerization level was decreased when the reaction was performed in a less-polar DMSO/toluene mixed solvent system (1.8% epi. level, entry 12; Table S1). Further investigation for N-hydroxy amide additives led us to identify that N-hydroxy-2-pyridinone methyl ester (HOPO$^{Me}$)[48,49] improved yield while reducing the epimerization level to <1% (entry 13). Increasing the amounts of alanine (2.0 equiv) and HOPO$^{Me}$ (2.0 equiv) enhanced the reactivity to give 86% yield of **2ba** after 6 hours (entry 14). However, separation of the crystalline HOPO$^{Me}$ from the tripeptide product was difficult. Further structural tuning afforded the optimal additive, HOPO$^{Phy}$ (entry 15), which could be easily recovered by a simple hexane washing and reused (Table S2).

**Scope and limitations**. After developing a practical isolation protocol of the products (recrystallization or flash chromatography, see the Method section), the optimized conditions were used to survey substrate generality (Fig. 2).

First, we investigated the generality of amino acids to be introduced using **1b** as a peptide substrate (Fig. 2a). The reaction proceeded in high yield (78–99%) with <1−1.8% epimerization for amino acids bearing hydrophobic (Val: **2bb**, Phe: **2bd**, Ile: **2be**, Met: **2bf**, Trp: **2bg**, $^t$Leu: **2bp**) or protected (Cys: **2bh**, Lys: **2bn**, Arg: **2bo**) side chains, due to their acceptable solubility in the optimized solvent. Specifically, the sterically hindered amino acid, $^t$Leu, was introduced to **1b** to produce tripeptide **2bp** containing a highly congested sequence (Phe-Val-$^t$Leu) in high yield (93%) with <1% epimerization. Ala and Gly were barely

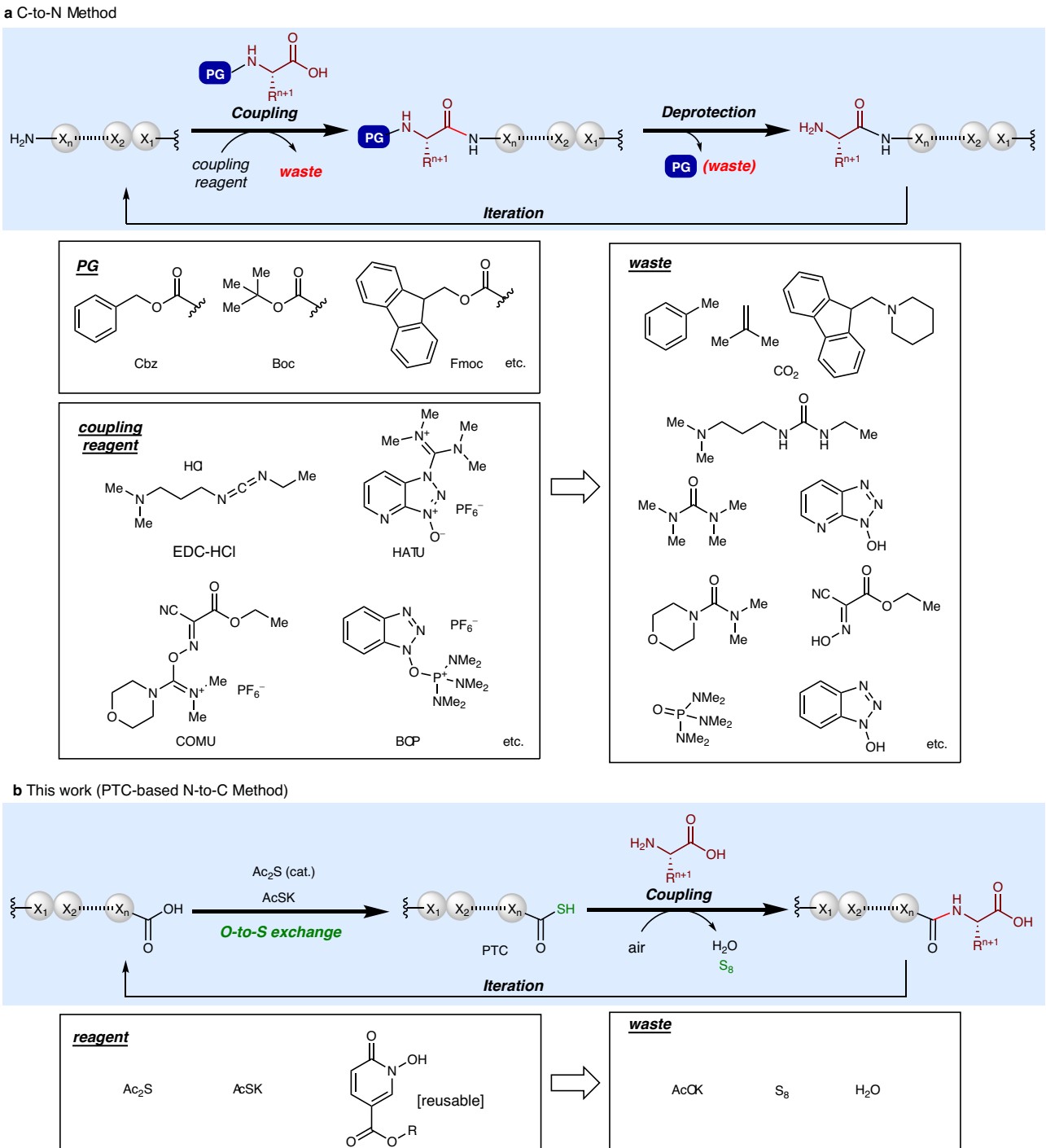

**a** C-to-N Method

**b** This work (PTC-based N-to-C Method)

**Fig. 1 C-to-N and N-to-C peptide syntheses. a** Traditional peptide synthesis elongates the peptide chain from C-terminus to N-terminus. Protecting groups (PG) at the N-terminus of the elongating amino acids and non-recoverable coupling reagents are necessary in excess amounts, leading to low atom and step efficiencies. **b** The N-to-C elongation developed in this work. Formation of peptide thiocarboxylic acid (PTC), followed by peptide coupling with non-protected amino acids in the presence of a reusable epimerization suppressor produces a peptide bond with minimal waste.

soluble in the solvent and produced slightly higher epimerization levels (ca. 2.1% for: **2ba** and 1.0% for **2bc**). For relatively less soluble amino acids bearing polar side chains (Thr, Tyr, Asp, Asn), however, the reaction under the above optimized conditions resulted in low yield (9–48%), likely due to insufficient concentration of the amino acids. In these cases, PTC hydrolysis preceded the desired peptide coupling. Further modifying the reaction conditions, we found that by using DMSO solvent

without added toluene, the HOPO$^{Me}$ additive which is more active than HOPO$^{Phy}$, a desiccant (MgSO$_4$), and/or microwave irradiation (40 °C, see section 1-1 in Supplementary Methods for detailed parameters), the desired tripeptides were obtained in high yield (67–94%) with <1% epimerization. Side chain protection was not necessary for amino acids containing functional groups of moderate nucleophilicity (Trp: **2bg**, Ser: **2bi**, Thr: **2bj**, Tyr: **2bk**, Asn: **2bl**, and Asp: **2bm**).

**Table 1 Optimization of PTC-based peptide synthesis.**

| Entry | PTC | Alanine (X equiv) | Additive | Oxidant | Solvent (Y mM) | Time (h) | Yield[a] (%) | epi. level[b] (%) |
|---|---|---|---|---|---|---|---|---|
| 1 | 1a | Ca(Ala)$_2$ (0.6) | none | FePc[c], air | DMF (10) | 3 | 22 | ND |
| 2 | 1a | Ca(Ala)$_2$ (0.6) | HOBt | FePc[c], air | DMF (10) | 3 | 44 | 2.3 |
| 3 | 1a | Ca(Ala)$_2$ (0.6) | HOAt | FePc[c], air | DMF (10) | 3 | 39 | 1.2 |
| 4 | 1a | Ca(Ala)$_2$ (0.6) | HOSu | FePc[c], air | DMF (10) | 3 | 20 | ND |
| 5 | 1a | Ca(Ala)$_2$ (0.6) | NHPI | FePc[c], air | DMF (10) | 3 | 16 | ND |
| 6 | 1a | Ca(Ala)$_2$ (0.6) | HOObt | FePc[c], air | DMF (10) | 3 | 33 | <1 |
| 7 | 1a | Ca(Ala)$_2$ (0.6) | HOObt | FePc[c], air | DMF (100) | 3 | 68 | <1 |
| 8 | 1b | Ca(Ala)$_2$ (0.6) | HOObt | FePc[c], air | DMF (100) | 3 | 35 | 6.1 |
| 9 | 1b | Ca(Ala)$_2$ (0.6) | HOObt | Air | DMF (100) | 22 | 64 | 4.9 |
| 10 | 1b | H-Ala-OH (1.2) | HOObt | Air | DMF (100) | 22 | 55 | 1.3 |
| 11 | 1b | H-Ala-OH (1.2) | HOObt | Air | DMSO (100) | 22 | 70 | 4.5 |
| 12 | 1b | H-Ala-OH (1.2) | HOObt | Air | DMSO/tol (1:1) (100) | 22 | 86 | 1.8 |
| 13 | 1b | H-Ala-OH (1.2) | HOPO$^{Me}$ | Air | DMSO/tol (1:1) (100) | 6 | 80 | <1 |
| 14[d] | 1b | H-Ala-OH (2.0) | HOPO$^{Me}$ | Air | DMSO/tol (1:1) (100) | 6 | 86 | <1 |
| 15[d] | 1b | H-Ala-OH (2.0) | HOPO$^{Phy}$ | Air | DMSO/tol (1:1) (100) | 6 | 78 | <1 |

ND not determined.
[a]Yield was determined by HPLC using a calibration curve.
[b]Epimerization level was calculated from crude mixtures as described in section 2-2 of Supplementary Methods.
[c]1 mol% of iron(II) phthalocyanine (FePc) was used.
[d]2.0 equiv of additive was used.

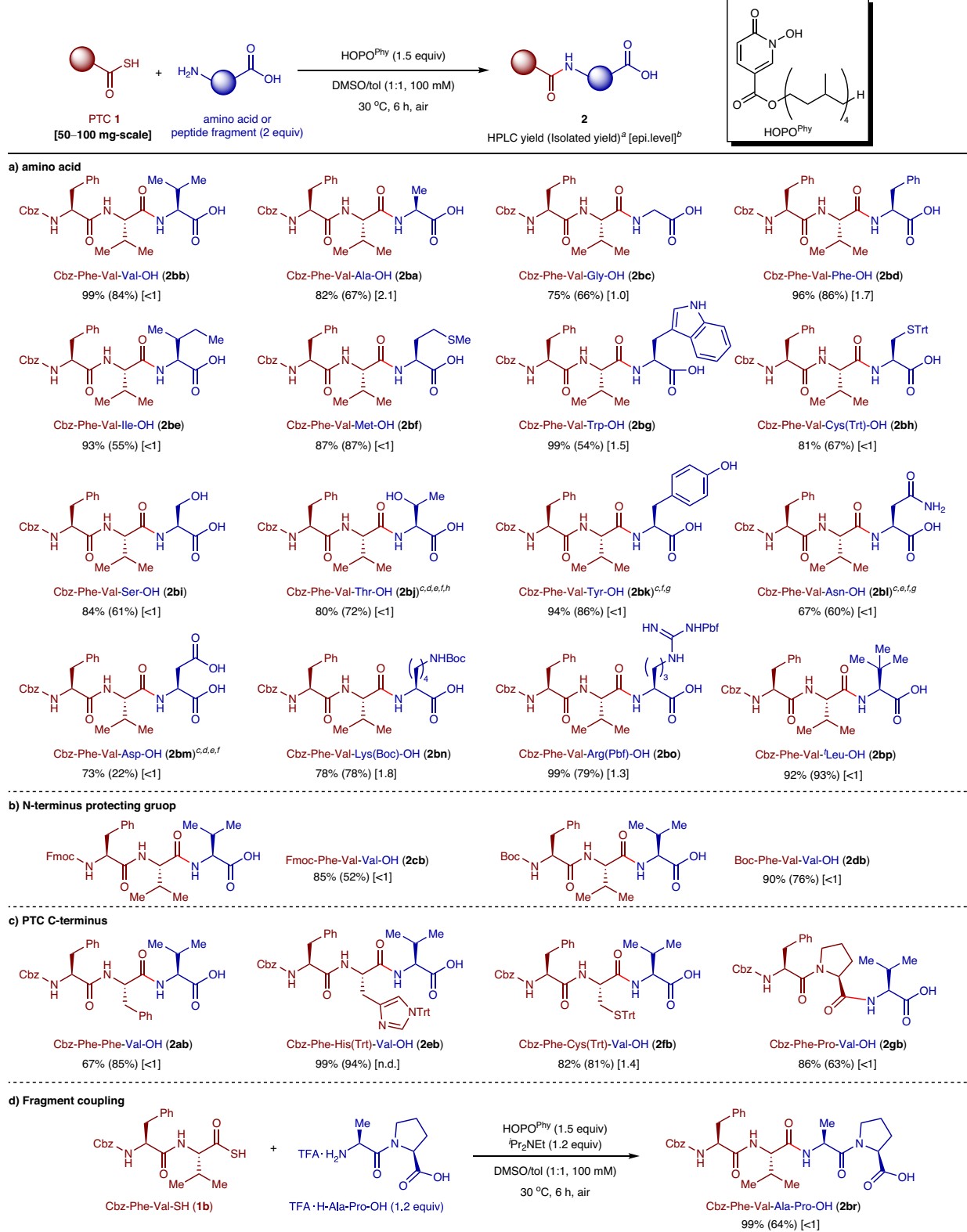

**Fig. 2 Substrate scope. a** Scope of main-chain unprotected amino acids to be introduced. **b** Scope of N-terminus protecting groups. **c** Scope of peptide C-terminus amino acids to be elongated. **d** Application to fragment coupling. [a]HPLC yields. The numbers in parentheses are the isolated yield of 50–100 mg-scale reaction. [b]Epimerization level was calculated from crude mixtures as shown in section 2-2 of Supplementary Methods. [c]DMSO (100 mM) was used as the solvent. [d] HOPO[Me] (3.0 equiv) was used instead of HOPO[Phy]. [e]MgSO$_4$ (2 g/mmol to **1**) was added. [f]Microwave irradiation at 40 °C. [g]HOPO[Phy] (3.0 equiv) was used. [h]Reaction time was 3 h. H-[t]Leu-OH = *tert*-leucine.

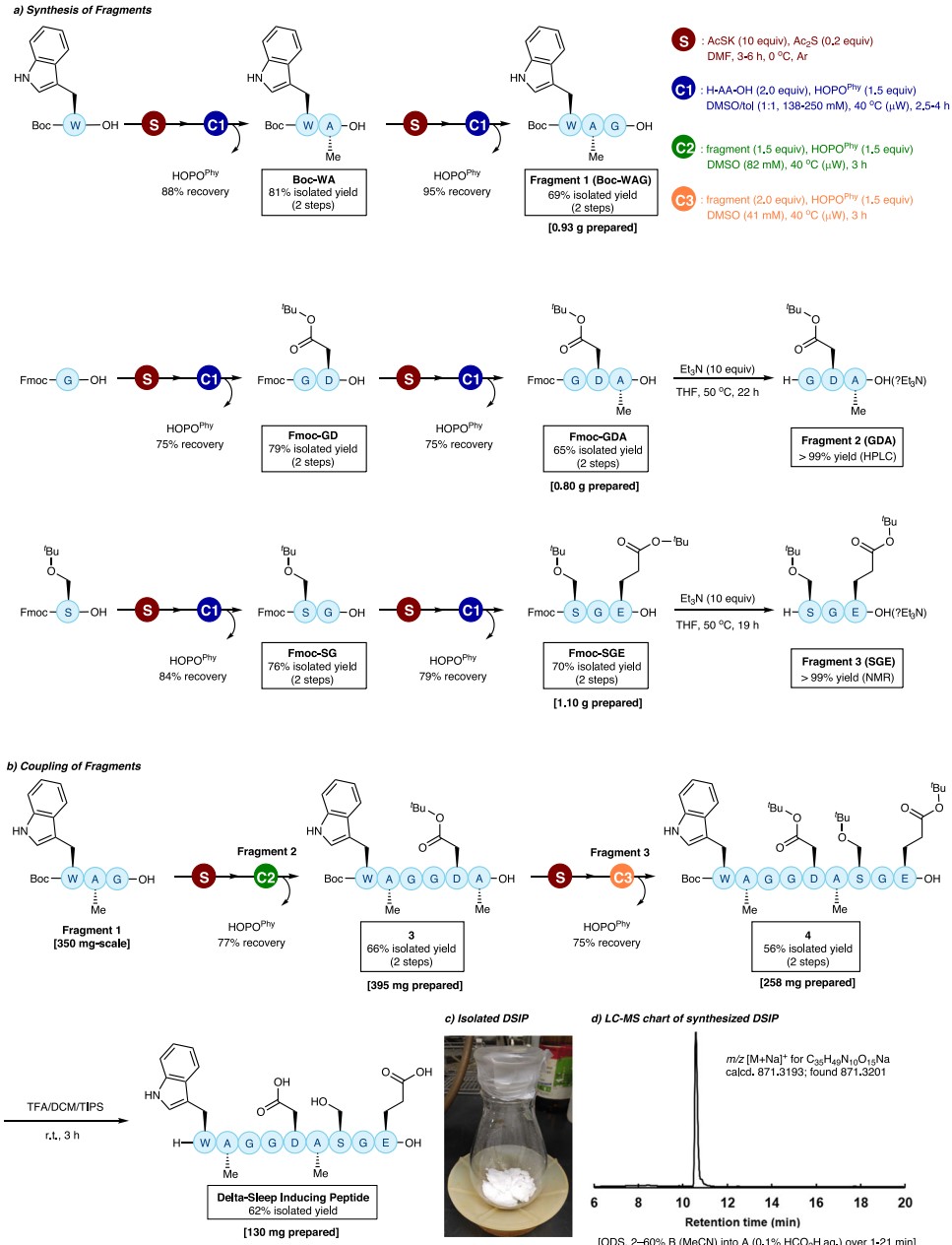

**Fig. 3 Scalable and convergent liquid-phase synthesis of DSIP. a** Synthesis of **Fragments 1–3** by iterative N-to-C peptide elongation. **b** Coupling of fragments and global deprotection leading to DSIP. **c** 130 mg of DSIP isolated as white powder. **d** Purity of DSIP was confirmed as >90% by LC-MS analysis.

Regarding the N-terminus protecting group on the PTC, Fmoc and Boc groups were also compatible with the current protocol (Fig. 2b: **2cb** and **2db**). As for the C-terminus amino acids of the elongating peptides, the reaction proceeded without any detectable epimerization at Phe (**2ab**) and Pro (**2gb**) residues (Fig. 2c). Furthermore, this method can be expanded to the convergent couplings of two peptide fragments. After liberation of the N-terminus amine from trifluoroacetic acid (TFA) salt of the peptide to be introduced with N,N-diisopropylethylamine ($^{i}$Pr$_2$NEt), the fragment coupling between dipeptides proceeded affording tetrapeptide **2br** in 99% yield without epimerization (Fig. 2d, see Fig. 3b for more examples of fragment coupling). Due to the higher solubility of peptide fragments compared to unprotected monoamino acids, only a slight excess (1.2 equiv) of C-terminus fragments was necessary. A preliminary application

of this method to solid-phase peptide synthesis (SPPS), however, resulted in only low-yield product formation (Table S3).

**Scalable, convergent liquid-phase synthesis of bioactive peptide**. We applied our method to the synthesis of a bioactive nonapeptide, delta-sleep-inducing peptide (DSIP) (Fig. 3). DSIP was retrosynthesized to three tripeptides, **Fragments 1–3**. As the starting amino acids for each fragment, we selected Boc-Trp for **Fragment 1** aiming at global deprotection under acidic conditions at the final step, and Fmoc-Gly and Fmoc-Ser for **Fragment 2** and **Fragment 3**, respectively, for selective deprotection prior to two fragment couplings.

After converting a C-terminus carboxylic acid to PTC, an unprotected amino acid was coupled under the conditions described above (Fig. 3a). The crude product obtained after

extraction with ethyl acetate (AcOEt) and evaporation of the solvent, was washed with hexane to extract HOPO$^{Phy}$ (75–95% recovery). The recovered and purified HOPO$^{Phy}$ was reusable without any loss of its activity. Then, the residue containing the product peptide was dissolved in AcOEt or MeOH, and the solution was treated with activated carbon. This process efficiently eliminated residual sulfur compounds, which were often problematic for the next peptide coupling and purification. The subsequent simple purification by silica gel column chromatography afforded the desired di- and tripeptides in good yield with sufficient purity for the next iteration or fragment coupling. For the removal of Fmoc group, we used triethylamine as a base. After evaporation, the crude mixture was dissolved in a biphasic solvent comprised of water and ether, which contained the product peptides and Fmoc-derived side products, respectively. The water phase was separated and freeze dried. Consequently, **Fragments 1–3** were synthesized in pure forms in a scalable manner (>300 mg prepared for each).

Then, fragment couplings were performed (Fig. 3b). After converting **Fragment 1** to PTC, the reaction with **Fragment 2** yielded hexapeptide **3** in 66% yield (2 steps) without column chromatography. Hexapeptide **3** was further converted to PTC and coupled with **Fragment 3** to yield protected DSIP **4** in 56% yield (2 steps) without column chromatography. Finally, global deprotection and purification by reverse-phase column chromatography afforded 130 mg of DSIP in 62% yield (Fig. 3c, d), showcasing that the current protocol is practical in supplying middle-sized bioactive peptides.

**Mechanistic studies**. To gain insight into the mechanism of the peptide coupling, the reaction was monitored over time by HPLC. When PTC **1b** was stirred under air without a coupling partner, **1b** was oxidatively dimerized to **5** within 1 h (Fig. 4a). Under the indicated conditions in the presence of HOPO$^{Phy}$, PTC **1b** was consumed in 1 h, producing tripeptide **2ba** in 81% yield. The formation of elemental sulfur (S$_8$) was confirmed by HPLC during the reaction. Dimer **5** was observed at the initial stage of

the reaction (t < 30 min) but was consumed within 3 h and converted to active ester **6**. Then, **6** gradually reacted with alanine and was fully converted to the tripeptide after 4.5 h (Fig. 4b, c). From these reaction profiles, the rate-limiting step is likely the peptide bond-forming step between active ester **6** and the amino acid (Figs. S1–S3).

Based on the above observations, a plausible reaction mechanism is proposed as shown in Fig. 5. First, PTC **1** dimerizes under aerobic conditions to form diacyl disulfide **7**. DMSO solvent accelerates this oxidation step[47]. Diacyl disulfide **7** then reacts with the additive HOPO$^R$ to produce active ester **8**, thus suppressing undesired epimerization through oxazolidine formation. The liberated acyl disulfide **9** reacts with **1** to generate **7** and H$_2$S, or with HOPO$^R$ to generate active ester **8** and H$_2$S$_2$. Active ester **8** gradually reacts with an unprotected amino acid or peptide fragment to produce elongated peptide **2**. H$_2$S and H$_2$S$_2$ undergo oxidation to release stable S$_8$ and water as the only byproducts.

## Conclusion

In this study, we developed an iterative, liquid-phase N-to-C peptide synthesis relying on the PTC platform, which enabled the use of unprotected amino acids as starting materials[50]. Only a one atom difference (O *vs.* S) distinguished the C-termini of elongating peptides and the unprotected amino acids being introduced. Therefore, protecting group manipulations and the number of synthetic steps were minimal compared with conventional C-to-N synthesis. A reusable additive (HOPO$^{Phy}$) bearing a long-alkyl chain allowed for coupling of amino acids or peptide fragments in high yield with minimal epimerization (<1% in most cases). The only waste byproducts were water and elemental sulfur, making the current process highly atom efficient. A straightforward workup (extraction, hexane washing, activated carbon treatment, and/or chromatography/recrystallization) after the coupling reaction provided peptides with sufficient purity for the next iteration. This method was applicable not only to sequential elongation of single amino acids but also to convergent

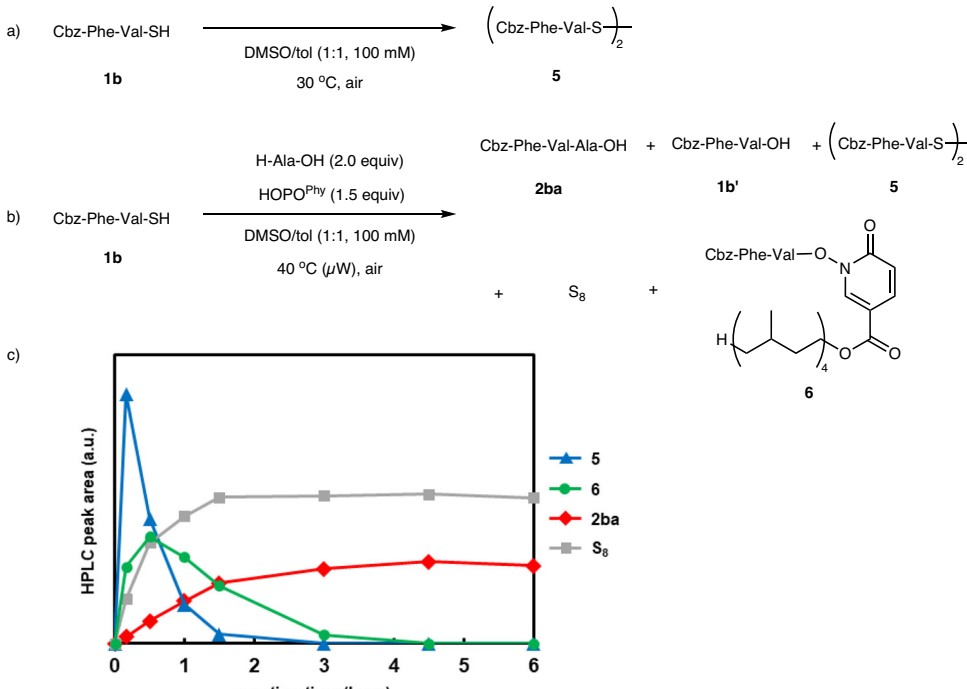

**Fig. 4 Reaction profile. a** Oxidative dimerization of PTC **1b** under air. **b** Analysis of reaction intermediates during the N-to-C peptide coupling between PTC **1b** and H-Ala-OH. **c** Time course tracking of the amounts of reaction intermediates by LC-MS.

**Fig. 5 Plausible reaction mechanism.** PTC **1** is oxidatively dimerized to produce diacyl disulfide **7**, which immediately forms active ester **8** by the reaction with HOPO$^R$, entering the N-to-C peptide coupling process to afford **2**.

fragment couplings, which allowed for the rapid increase in molecular complexity. Taking advantage of the characteristics of this method, a practical synthesis of a bioactive nonapeptide was demonstrated in a sub-gram scale. Although there are several previous examples of N-to-C peptide elongation, their scopes were limited or not thoroughly studied. Furthermore, these earlier works require protecting groups or activating groups at the N- or C-terminus, which diminished the potential advantages of N-to-C elongation regarding atom efficiency. Our achievement will open a green route to practically supplying peptides of ca. 10 residues in length, a common size for synthetic peptide drugs. Further studies investigating ways to accelerate the reaction rate while reducing epimerization, as well as the applicability to late-stage peptide functionalization, lateral coupling, cyclic peptide synthesis, and SPPS, are ongoing.

## Methods

**Procedure for gram-scale N-to-C peptide coupling (represented by the synthesis of Fragment 1).** To a solution of **Boc-WA** (1.18 g, 3.14 mmol) and potassium thioacetate (3.59 g, 31.4 mmol, 10 equiv) in DMF (31.4 mL), diacetyl sulfide (65.7 μL, 0.628 mmol, 0.2 equiv) was added dropwise at 0 °C and the mixture was stirred for 3.5 hours at 0 °C under an argon atmosphere. Ethyl acetate, water, and 1 M HCl aq. were added to the reaction mixture to stop the reaction. The products were extracted with ethyl acetate. The combined organic layers were washed with water, 1 M HCl aq., and brine, dried over Na$_2$SO$_4$, and filtered. Volatiles were removed under reduced pressure to afford the crude PTC (**Boc-WA-SH**). This crude product was used for the peptide coupling reaction without further purification.

**Boc-WA-SH.** (estimated as 3.14 mmol) dissolved in DMSO (6.5 mL) was added to a mixture of glycine (471 mg, 6.18 mmol, 2.0 equiv), HOPO$^{Phy}$ (2.05 g, 4.71 mmol, 1.5 equiv), and toluene (6.5 mL) in a test tube for a microwave apparatus. The mixture was stirred under microwave irradiation at 40 °C for 3 hours. Ethyl acetate, water, and 1 M HCl aq. were added to the reaction mixture. The mixture was extracted with ethyl acetate. The combined organic layers were washed with brine, dried over Na$_2$SO$_4$, filtered, and volatiles were removed under reduced pressure. Hexane (150 mL) was added to the residue and the mixture was sonicated to precipitate out the peptide product. The precipitates were collected by filtration and washed with hexane. Then, the filtrate was evaporated under reduced pressure for recovering HOPO$^{Phy}$. The resulting residue from hexane was purified by column chromatography (neutral silica gel, hexane/ethyl acetate = 80:20 → 50:50) to afford HOPO$^{Phy}$, which was reusable for another peptide coupling reaction (1.95 g, 95% recovery).

Meanwhile, the sticky precipitate on the filter containing tripeptide was once dissolved into a large amount of methanol and ethyl acetate. Then, the solvent was removed under reduced pressure. Ethyl acetate (50 mL) and activated carbon (400 mg) were added to the mixture. This suspension was stirred at 80 °C for 10 min under an argon atmosphere and then cooled to room temperature. Activated carbon was filtered over Celite and washed with ethyl acetate. The filtrate was evaporated under reduced pressure to afford the tripeptide product, **Fragment 1**. This crude product was purified by column chromatography (silica gel, hexane/ethyl acetate = 70:30 → 0:100 then ethyl acetate/methanol = 90:10 → 80:20) to afford pure **Fragment 1** (931 mg, 69%).

**Procedure for peptide fragment coupling (represented by the synthesis of 2br).** To a solution containing Cbz-Phe-Val-SH (**1b**, 50 mg, 0.12 mmol), H-Ala-Pro-OH TFA salt (43 mg, 0.144 mmol, 1.2 equiv), and HOPO$^{Phy}$ (78.4 mg, 0.18 mmol, 1.5 equiv) in DMSO (600 μL) and toluene (600 μL), $^i$Pr$_2$NEt (25 μL, 0.144 mmol, 1.2 equiv) was added. After stirring at 30 °C for 6 hours, a HPLC sample was prepared (12 μL of the reaction mixture was picked up into 68 μL of 1% TFA/DMSO) for yield determination. HPLC yield was determined as 99% (method B in ESI).

TFA (68 μL) was added to the reaction mixture to quench the reaction. After transferring the reaction mixture into a separatory funnel, ethyl acetate and 1 M HCl aq. were added. Organic compounds were extracted with ethyl acetate (three times). The combined organic layers were washed with brine and dried over Na$_2$SO$_4$. After filtration, volatiles were removed under reduced pressure. The crude mixture was purified by column chromatography (silica gel, hexane/ethyl acetate = 80:20, then chloroform/methanol = 100:0 → 80:20). The obtained material was further purified by reverse-phase preparative HPLC (method D in section 1-4 of Supplementary Methods, t$_R$ = 52.0 min). Fractions containing the pure desired product were combined and lyophilized to afford analytically pure **2br** (43.3 mg, 64% isolated yield). For NMR data of isolated new compounds, see Supplementary Data 1.

## Data availability

All relevant data are presented in the main article or the supporting information. Detailed experimental procedures for the syntheses and characterizations of new compounds, mechanistic studies, and HPLC analysis are available in Electronic Supplementary Information. $^1$H and $^{13}$C NMR charts of isolated new compounds can be found in the Supplementary Data 1.

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

# Acknowledgements

This research was supported partly by MEXT/JSPS KAKENHI Grant number JP23H05466 and 23H04909 (M.K.) and JP21H05077 (K. Oisaki), and Inamori Research Grant (to K. Oisaki). T.T. thanks JSPS Fellowship JP2101365 for financial support. T.M. thanks Graduate Program for Leaders in Life Innovation (GPLLI). We thank Professor Akira Otaka in Tokushima University for valuable suggestions.

# Author contributions

T.T., K.S., T.M., R.H., K. Oikawa, and M.N. conducted experimental studies and collected the data. K. Oisaki and M. K. designed, advised, and directed the project. M.Y., K. Oisaki and M.K. co-wrote the manuscript. All the authors analyzed the data, discussed the results, and edited the manuscript.

# Competing interests

The authors declare no competing interests.

## Additional information

**Peer review information** : *Communications Chemistry* thanks Rita Petracca and the other, anonymous, reviewers for their contribution to the peer review of this work. A peer review file is available.

