## [Peer Review File · Communications Chemistry]

Reviewers' comments:

Reviewer #1 (Remarks to the Author):

In this article, the authors efficiently describe a novel N-to-C synthetic method to achieve linear peptides with high yield and purity in an environmentally-friendly fashion. The large chemical waste, associated to the standard peptide C-to-N chain elongation, is prevented by using unprotected amino acid building blocks and only one recyclable N-hydroxy pyridone additive to promote condensation and suppress epimerization. The new synthetic method, together with the hypothesis of the reaction mechanism, is extensively discussed and the scope of the reaction is efficiently reported. As a proof of concept for their newly developed elongation strategy, the straightforward synthesis of the nonapeptide DSIP is described. The materials and methods utilised for the work are described in detail and the supporting information document contains all necessary scientific data to support the results presented in the main article. The authors clearly achieved what they claim, and they seem aware of the limitations that the method may still have, mainly concerning the moderate-long times of reaction and the epimerisation still happening in low percentage. Overall, the work described is novel and of high impact in the peptide field and in general in organic synthesis; it will undoubtedly prompt researchers in the peptide field to explore new synthetic routes and working on finding out more efficient and greener options for peptide synthesis. I believe that the paper is also worth to be highlighted in News and Views of the Nature journal. I would be interested and happy to write a piece about it.

Recommendation: accept for publication with minor comments.

Minor points and questions to the authors:

1. With your methodology you will reduce massively the amount of waste usually associated to peptide synthesis. This is relevant for the environment and rightfully highlighted in the text. You are still proposing the use of toluene as co-solvent. Did you try to conduct the reaction in less toxic solvents? DMSO is also not an ideal choice of solvent in liquid organic synthesis. Do you believe it is necessary to promote the thioacids oxidation?
2. Concerning the synthesis of amino-thioacids. You do not characterise the compounds but use them directly in the next coupling step. How are you sure of a full conversion of the amino-acid to amino-thioacid? I had experience in working with thioacids and they are highly prone to hydrolysis. Do you observe this? Did you think of preparing protected versions of thioacids? You can look at this paper <https://doi.org/10.1002/ejoc.202100615>
3. Could your reaction be conducted on solid phase? In this case, would you immobilise your N terminus amino-acid on it? Can the thioacid still be formed? I believe that if your method is only feasible for liquid phase synthesis it needs to be reported in the main text.
4. For the synthesis of the final nonapeptide, why do you prepare three fragments? Did you try to perform the synthesis via single amino-acid addition? Is it to avoid higher percentage of epimerisation? What happens if you want to make longer peptides?
5. Did you consider late-stage and/or selective lateral chain functionalisation? I see more disadvantages in using your methodology for these compared to the existing approaches. And what about cyclisation? Could you implement this step in your strategy? Worth adding comments related to these topics in the main text.

Reviewer #2 (Remarks to the Author):

Referee Report on "Protecting Group Minimum, Practical N-to-C Peptide Synthesis" by M. Kanai et al. The manuscript submitted by M. Kanai and colleagues titled "Protecting Group Minimum, Practical N-to-C Peptide Synthesis" presents an iterative N-to-C peptide elongation method, deviating from the conventional C-to-N bond formations predominant in the field. The authors employ a two-step procedure involving the activation of carboxylic acid as a thiocarboxylic acid using an excess of potassium thioacetate and thioacetic anhydride catalysis. Subsequently, the amide link is formed with an unprotected α -amino acid in the presence of a recoverable racemization-suppressing reagent. The manuscript demonstrates high-quality writing and merits publication. However, it raises some concerns that prevent the acceptance for a publication in Communications Chemistry. Liquid phase peptide synthesis is secondary to the solid phase counterpart (SPPS), which allows peptides with up to 50 amino acid residues as mentioned in the introduction. Probably the present method may not be suitable for SPPS due to potential difficulties in forming diacyl disulfide using solid phase strategy. This point must be clarified and is somewhat misleading in the introductory part. Furthermore, upon reviewing a previous contribution from the same group ("A Catalytic One-Step Synthesis of Peptide Thioacids: The Synthesis of Leuprorelin via Iterative Peptide–Fragment Coupling Reactions. Chem. Commun. 2018, 54 (86), 12222–12225. <https://doi.org/10.1039/C8CC07935H>"), it becomes evident that essentially the same strategy was already reported by the authors, and a similar approach was also documented in the work of S. J. Danishefsky (J. Am. Chem. Soc. 2010, 132 (47), 17045–17051. <https://doi.org/10.1021/ja1084628>). Consequently, the novelty of the present work mainly lies in the use of a recoverable HOPO racemization-suppressing reagent, and the results may have been somewhat overstated. Therefore, I suggest that this work is more appropriately suited for publication in a more specialized journal than Communication Chemistry.

Additional comments:

1. It would be helpful to provide information regarding the treatment of the "air" used, as the authors mentioned that thioacids may undergo hydrolysis. Consider experimenting with pure dry oxygen instead of air to accelerate and control the water content of the coupling reactions. Additionally, the use of a metal catalyst could be explored to further aid in this direction.
2. The use of functionalized amino acids (2bj-2bm) requires modified conditions, including microwave conditions. The manuscript should accurately report the reaction conditions. Details like "Stirred in a microwave apparatus [Biotage Initiator+] at 40 °C for 3 hours" are insufficient. It is crucial to mention the level of irradiation maintained and whether a cooling system was used to control the temperature, considering that rapid temperature rises may occur in the presence of polar DMSO. If the temperature was controlled solely by irradiation, it should be clarified that very low levels of irradiation were employed intermittently, and the reaction was conducted under primarily thermal conditions.
3. A specific reference concerning HOPOMe and HOPOPhy would be of some good use for the reader.

In summary, while the work presented in this manuscript showcases a unique approach to N-to-C peptide elongation, its novelty may be limited by previous reports from the same authors and related

works in the field. Therefore, I recommend addressing the concerns raised and considering submission to a more specialized journal that would better appreciate the contribution of the recoverable HOPO racemization-suppressing reagent.

Reviewer #3 (Remarks to the Author):

The work outlined in the manuscript details a bright strategy for the synthesis of peptides through an N to C strategy. The active species is diacyldisulfide that in the presence of a derivative of HOPO renders the active ester.

The HOPO derivative can be recovered due to the presence of an aliphatic tail.

The work is really well done and the results are supported by the analytical data.

Have the authors identified the epimers in the crude of the model peptide synthesized?

Just a few suggestions, queries

The title is a little bit misleading

“Protecting Group-Minimum, Practical N-to-C Peptide Synthesis”

I would like to suggest,

“Practical N-to-C Peptide Synthesis”

What means “drug modalities”?

The following sentence “Because the two amino groups, one in the elongating peptide strand and the other in the amino acid to be introduced, are already differentiated as amide and amine groups, respectively, this strategy is potentially advantageous in improving both atom and step efficiency by minimizing protecting group manipulations” also applied to the C to N, one is an amine other is a carbamate.

Responses to Reviewer 1's Comments:

1. *With your methodology you will reduce massively the amount of waste usually associated to peptide synthesis. This is relevant for the environment and rightfully highlighted in the text. You are still proposing the use of toluene as co-solvent. Did you try to conduct the reaction in less toxic solvents? DMSO is also not an ideal choice of solvent in liquid organic synthesis. Do you believe is necessary to promote the thioacids oxidation?*

According to the reviewer's comment, we studied greener solvents than toluene, combined with DMSO. As the reviewer pointed out, and as was described in the main text, DMSO promoted thioacid oxidation, and thus was required for the practical reaction kinetics and yield. AcOEt or AcOⁱPr produced a slightly lower yield than toluene. Cyclopentyl methyl ether produced a slightly higher yield and epimerization level than toluene. Increasing the DMSO/toluene ratio from 1:1 to 10:1 produced a significantly higher epimerization level. Therefore, DMSO/toluene 1:1, used in the original submission, is the optimized solvent system. The results of the solvent screening have been described in Section 3-3 in ESI.

2. *Concerning the synthesis of amino-thioacids. You do not characterise the compounds but use them directly in the next coupling step. How are you sure of a full conversion of the amino-acid to amino-thioacid? I had experience in working with thioacids and they are highly prone to hydrolysis. Do you observe this? Did you think of preparing protected versions of thioacids? You can look at this paper <https://doi.org/10.1002/ejoc.202100615>*

We synthesized peptide thiocarboxylic acids (PTCs) according to our previous report (ref 44) with slight modifications (please see Section 4-1 in ESI). For every PTC, we checked the conversion and purity by HPLC, both of which were generally high (generally >90% conversion and >95% purity, despite moderate isolated yield due to instability of PTC; e.g., 99% conversion for Boc-WA-SH shown in Section 5-1 in ESI). Some PTCs were purified by silica gel column chromatography despite that partial decomposition proceeded during the purification (please see Section 4-1 in ESI and ref 44). Furthermore, the successful synthesis of nonapeptide DISP in high purity and yield (Figure 2) demonstrated the practical efficiency of the overall process, including multiple steps for the PTC synthesis. To show the high efficiency of the PTC synthesis, we have included HPLC charts of the reaction mixture and after isolation for each PTC synthesis in Section 10-1 in ESI.

As for the sensitivity of PTC toward hydrolysis, we evaluated the time-dependent hydrolysis of PTC **1b** in a CH₃CN/H₂O (3:1) solvent at room temperature, by HPLC. **1b** was rather stable; 6% and 10% hydrolyses were observed in 6 h and 12.5 h, respectively. We have included these results in Section 8 in ESI.

Thank you for letting us know a very intriguing paper related to PTC. Because the use of protecting groups increases the number of synthetic steps and waste, we did not intend to employ protected PTCs.

3. *Could your reaction be conducted on solid phase? In this case, would you immobilise your N terminus amino-acid on it? Can the thioacid still be formed? I believe that if your method is only feasible for liquid phase synthesis it needs to be reported in the main text.*

Following the suggestion of this reviewer, we examined the solid-phase PTC-based N-to-C peptide synthesis using

inverse 2-Cl trityl resin protocol (F. Albericio and co-workers, *Org. Lett.* **2000**, *2*, 1815). The PTC synthesis and condensation with benzylamine, which has less solubility concern, were attempted as shown below. The desired amide was produced only in low yield (5–8%) and starting phenylalanine was majorly recovered after cleavage from the resin under acidic conditions. We speculate two main reasons for the low yield of the on-resin reaction; PTC synthesis was not sufficient and oxidative dimerization hardly proceeded. Therefore, the application of the current method to the solid-phase synthesis requires further intensive studies.

Entry	Ac ₂ S	AcSK	Yield
1	1	10	5
2	3	30	8

Yield was determined by LCMS.

Accordingly, we have included the above results in Section 9 in ESI, and the following sentence at the end of “Scope & Limitations” in the main text; A preliminary application of this method to solid-phase peptide synthesis (SPPS), however, resulted in only low-yield product formation (see ESI). We also stated in “Conclusion” as; Further studies investigating ways to accelerate the reaction rate while reducing epimerization, as well as the applicability to late-stage peptide functionalization, lateral coupling, cyclic peptide synthesis, and SPPS, are ongoing.

4. For the synthesis of the final nonapeptide, why do you prepare three fragments? Did you try to perform the synthesis via single amino-acid addition? Is it to avoid higher percentage of epimerisation? What happen if you want to make longer peptides?

The peptide fragment coupling rapidly and convergently increases molecular complexity from relatively short peptide segments. In this sense, the fragment coupling is advanced to the stepwise coupling. It is highly advantageous if a single method can promote both fragment and stepwise couplings. We used the combination of fragment and stepwise couplings in the DSIP synthesis to demonstrate the favorable characteristics of our method.

Based on the reviewer’s comment, we modified a sentence in “Conclusion” in the main text, as follows; This method was applicable not only to sequential elongation of monoamino acids but also to convergent fragment couplings, which allowed for the rapid increase in molecular complexity.

5. *Did you consider late-stage and/or selective lateral chain functionalisation? I see more disadvantages in using your methodology for these compared to the existing approaches. And what about cyclisation? Could you implement this step in your strategy? Worth adding comments related to these topics in the main text.*

Thank you for your insightful comment. We believe that the fragment coupling between hexapeptide **3** and tripeptide **Fragment 3** in our DSIP synthesis (Figure 2) is an example of late-stage functionalization, showcasing the tolerance of our method to free indole, ester, ether, and amide functionalities. According to the reviewer's comment, however, we would like to clarify functional group compatibility in more detail and investigate a feasibility for lateral chain functionalization and cyclization, in our future studies. We have included those future perspectives in the final sentence in "Conclusion", as follows; Further studies investigating ways to accelerate the reaction rate while reducing epimerization, as well as the applicability to late-stage peptide functionalization, lateral coupling, cyclic peptide synthesis, and SPPS, are ongoing.

Responses to Reviewer #2:

1. *Liquid phase peptide synthesis is secondary to the solid phase counterpart (SPPS), which allows peptides with up to 50 amino acid residues as mentioned in the introduction. Probably the present method may not be suitable for SPPS due to potential difficulties in forming diacyl disulfide using solid phase strategy. This point must be clarified and is somewhat misleading in the introductory part.*

We understand the high utility of SPPS. We believe that the introductory part correctly appreciates SPPS. Liquid phase peptide synthesis is still important especially for scalable peptide supply.

According to the reviewer's comment, we examined the solid-phase PTC-based N-to-C peptide synthesis using inverse 2-Cl trityl resin protocol (F. Albericio and co-workers, *Org. Lett.* **2000**, 2, 1815). The PTC synthesis and condensation with benzylamine, which has less solubility concern, were attempted as shown below. The desired amide was produced only in low yield (5–8%) and starting phenylalanine was majorly recovered after cleavage from the resin under acidic conditions. We speculate two main reasons for the low yield of the on-resin reaction; PTC synthesis was not sufficient and oxidative dimerization hardly proceeded. Therefore, the application of the current method to the solid-phase synthesis requires further intensive studies.

Entry	Ac ₂ S	AcSK	Yield
1	1	10	5
2	3	30	8

Yield was determined by LCMS.

Accordingly, to avoid misleading, we have repetitively described in “Introduction” that the method developed in this study is a liquid-phase peptide synthesis. Further, we have included the above results in Section 9 in ESI, and the following sentence at the end of “Scope & Limitations” in the main text; A preliminary application of this method to solid-phase peptide synthesis (SPPS), however, resulted in only low-yield product formation (see ESI). We also stated in “Conclusion” as; Further studies investigating ways to accelerate the reaction rate while reducing epimerization, as well as the applicability to late-stage peptide functionalization, lateral coupling, cyclic peptide synthesis, and SPPS, are ongoing.

- Furthermore, upon reviewing a previous contribution from the same group (“A Catalytic One-Step Synthesis of Peptide Thioacids: The Synthesis of Leuprorelin via Iterative Peptide–Fragment Coupling Reactions. *Chem. Commun.* 2018, 54 (86), 12222–12225. <https://doi.org/10.1039/C8CC07935H>”), it becomes evident that essentially the same strategy was already reported by the authors, and a similar approach was also documented in the work of S. J. Danishefsky (*J. Am. Chem. Soc.* 2010, 132 (47), 17045–17051. <https://doi.org/10.1021/ja1084628>). Consequently, the novelty of the present work mainly lies in the use of a recoverable HOPO racemization-suppressing reagent, and the results may have been somewhat overstated. Therefore, I suggest that this work is more appropriately suited for publication in a more specialized journal than *Communication Chemistry*.

As pointed out by the reviewer and also described in the main text, we used the previously reported method by our group (ref 44) for the synthesis of peptide thioacids (PTC). The main issue of this manuscript is not the PTC synthesis, but is the peptide coupling between PTCs and main-chain unprotected amino acids/peptides. In ref 44, we reported a peptide (Leuprorelin) synthesis using PTCs; however, this relied on a known peptide coupling method using an N-2,4-dinitrobenzenesulfonyl protecting group, which is distinct from the peptide coupling method developed in our present work.

Our peptide coupling method has been developed inspired by previous reports (refs 34–43), including Danishefsky's work (ref 37), as the reviewer commented. However, Danishefsky's work was a peptide fragment coupling, not a sequential elongation. Furthermore, C-terminus protected peptides were used, the reaction time was long (12–48 h), and 3–6% epimerization was often observed. Thus, our method is the first peptide coupling using main-chain unprotected amino acid/peptide substrates and applicable to both iterative elongation and fragment coupling completed within a short reaction time (3–6 h), promising a practical liquid-phase peptide synthesis with markedly reducing the number of synthetic steps and waste, compared to previous methods.

We described the limitations of previous PTC couplings in the first paragraph of the "Optimization of Conditions" section and novelties of our method in "Conclusion", in our original submission. To clarify the novelty of our method further, we have modified a sentence in "Conclusion", as follows; Therefore, protecting group manipulations and the number of synthetic steps were minimal compared with conventional C-to-N synthesis.

Additional comments:

- 3. It would be helpful to provide information regarding the treatment of the "air" used, as the authors mentioned that thioacids may undergo hydrolysis. Consider experimenting with pure dry oxygen instead of air to accelerate and control the water content of the coupling reactions. Additionally, the use of a metal catalyst could be explored to further aid in this direction.*

Thank you for your insightful comment. According to the reviewer's comment, we monitored the reaction either under air or O₂ atmosphere using HPLC and found that the formation or consumption of the product or intermediate was comparable (See Section 7 in ESI). Therefore, the concentration of O₂ does not significantly affect the reaction kinetics.

The hydrolytic resistance of PTC was also evaluated in an acetonitrile/water (3/1) solvent, and hydrolysis was observed only little within the reaction time (<6% hydrolysis in 6 h, see Section 8 in ESI). This result indicates that water generated in the reaction mixture is not problematic.

In a primary phase of our study, we surveyed metal catalysts other than iron phthalocyanine (Table 1). However, we did not observe any positive effects. Furthermore, it was difficult to remove the metal catalysts from the product. Therefore, we did not pursue the use of metal catalysts or complexes further.

- 4. The use of functionalized amino acids (2bj-2bm) requires modified conditions, including microwave conditions. The manuscript should accurately report the reaction conditions. Details like "Stirred in a microwave apparatus [Biotage Initiator+] at 40 °C for 3 hours" are insufficient. It is crucial to mention the level of irradiation maintained and whether a cooling system was used to control the temperature, considering that rapid temperature rises may occur in the presence of polar DMSO. If the temperature was controlled solely by irradiation, it should be clarified that very low levels of irradiation were employed intermittently, and the reaction was conducted under primarily thermal conditions.*

We appreciate for the reviewer's comment. The temperature was controlled by intermittently irradiating low-level microwave, and the reaction was conducted under primarily thermal conditions. According to the reviewer's comment, we have included detailed microwave conditions in the 1-1. "General" section in ESI.

5. *A specific reference concerning HOPO^{Me} and HOPO^{Phy} would be of some good use for the reader.*

As the reviewer commented, we have added references 48 and 49 for HOPO^{Me} in the main text. HOPO^{Phy} is a new compound. Its synthesis and spectrum data were detailed in ESI.

6. *In summary, while the work presented in this manuscript showcases a unique approach to N-to-C peptide elongation, its novelty may be limited by previous reports from the same authors and related works in the field. Therefore, I recommend addressing the concerns raised and considering submission to a more specialized journal that would better appreciate the contribution of the recoverable HOPO racemization-suppressing reagent.*

Please see our response to comment 2 from the reviewer.

Responses to Reviewer #3:

1. *Have the authors identified the epimers in the crude of the model peptide synthesized?*

For identification of epimers, we synthesized authentic samples and compared the retention times by HPLC. To clarify this point, we have included the following sentence in Section 10-2 “Oligopeptide products 2” of ESI; We synthesized authentic epimer samples and assigned the HPLC peaks by comparing the retention times. The assignments of HPLC peaks other than the product and epimer were shown in Section 6.

Just a few suggestion, queries

2. *The title is a little bit misleading “Protecting Group-Minimum, Practical N-to-C Peptide Synthesis”. I would like to suggest, “Practical N-to-C Peptide Synthesis.*

Thank you for your suggestion. The applicability to main chain unprotected amino acids and peptides is a key feature of our method. Therefore, we have decided to retain the title as it is.

3. *What means “drug modalities”?*

It is a well-accepted term indicating approaches or strategies for the drug development. Therefore, we retained the term as it is.

4. *The following sentence “Because the two amino groups, one in the elongating peptide strand and the other in the amino acid to be introduced, are already differentiated as amide and amine groups, respectively, this strategy is potentially advantageous in improving both atom and step efficiency by minimizing protecting group manipulations” also applied to the C to N, one is an amine other is a carbamate.*

The use of protecting groups is inevitable with carbamates, which contradicts to our intended approach.

REVIEWERS' COMMENTS:

Reviewer #1 (Remarks to the Author):

The authors assessed all the comments in a detailed and critical way. The main text was changed accordingly. More experiments were performed to demonstrate their theories and rightfully answer the given questions. The overall revised manuscript is satisfactory and ready for publication.

Reviewer #1's comment:

The authors assessed all the comments in a detailed and critical way. The main text was changed accordingly. More experiments were performed to demonstrate their theories and rightfully answer the given questions. The overall revised manuscript is satisfactory and ready for publication.

Our response:

Thank you very much for your acceptance of our revised manuscript.